# Direct Utilization of Near-Infrared Light for Photooxidation with a Metal-Free Photocatalyst

**DOI:** 10.3390/molecules27134047

**Published:** 2022-06-23

**Authors:** Le Zeng, Zhonghe Wang, Tiexin Zhang, Chunying Duan

**Affiliations:** State Key Laboratory of Fine Chemicals, Dalian University of Technology, Dalian 116024, China; zengle@dlut.edu.cn (L.Z.); zhhwang@mail.dlut.edu.cn (Z.W.)

**Keywords:** near-infrared light, BODIPY, photoredox catalysis, prodrug activation

## Abstract

Near-infrared (NIR) light-triggered photoredox catalysis is highly desirable because NIR light occupies almost 50% of solar energy and possesses excellent penetrating power in various media. Herein we utilize a metal-free boron dipyrromethene (BODIPY) derivative as the photocatalyst to achieve NIR light (720 nm LED)–driven oxidation of benzylamine derivatives, sulfides, and aryl boronic acids. Compared to blue light–driven photooxidation using Ru(bpy)_3_Cl_2_ as a photocatalyst, NIR light–driven photooxidation exhibited solvent independence and superior performance in large-volume (20 mL) reaction, presumably thanks to the neutral structure of a BODIPY photocatalyst and the deeper penetration depth of NIR light. We further demonstrate the application of this metal-free NIR photooxidation to prodrug activation and combination with Cu-catalysis for cross coupling reaction, exhibiting the potential of metal-free NIR photooxidation as a toolbox for organic synthesis and drug development.

## 1. Introduction

In the past decades, photoredox catalysis has undergone unprecedented growth and become an important tool for organic synthesis, drug development, and polymer science [1,2]. However, the applied incident light is mainly ultraviolet or short wavelength visible light (*λ* < 500 nm) in the current photocatalysis setup, which leads to challenges and limitations [3,4]. For example, the short-wavelength incident light will not be exclusively absorbed by the photocatalyst in the presence of the colored reagents or reaction intermediates, leading to the formation of by-products, to low product yield, and to limited reaction scope. In addition, due to the shallow penetration of short wavelength light in various reaction media [5], the scale-up of visible light–driven photocatalysis suffers from slow reaction rates and decreased reaction yields, which is detrimental to industrial application.

In this context, utilizing a longer-wavelength light, in particular near-infrared (NIR, *λ* > 650 nm) light, as an energy source for photocatalysis emerged as a hot topic since it can obviate the above-mentioned limitations [5,6,7]. Compared to visible light, NIR light exhibits higher penetration depth in various media with weak scattering and diffuse reflection, especially for biological tissue, thus benefiting the effective light absorption by a photocatalyst in scale-up reactions [5,8,9]. The weaker energy of NIR light compared to visible light can circumvent light-induced degradation of the photocatalyst and substrates, thus enabling lower catalyst loading and better reaction selectivity [5,8,9]. In addition, the nearly 50% occupancy of NIR in solar energy provides a clean and infinite source of NIR light [8]. Despite these obvious advantages, NIR light–driven photoredox catalysis was rarely reported [8,9,10], largely due to the poor absorption of NIR light for conventional photocatalysts [3,4]. Currently there are two different approaches to achieve NIR photoredox catalysis. One is the indirect utilization of NIR light via an up-conversion strategy that converts NIR light to high-energy visible light; for example, the triplet–triplet annihilation up-conversion (TTA-UC) strategy [11,12]. Normally, the photoactivation procedure of TTA-UC–mediated NIR photoredox catalysis requires the involvement of multiple energy transfer steps between the sensitizer, annihilator, and visible light photocatalyst [11]. Obviously, the complexity makes this indirect approach not easy to handle. On the other hand, the direct utilization of NIR for photoredox catalysis relies on the use of a noble metal–based photocatalyst, which is expensive and might lead to toxic heavy-metal residue for the product [8,9]. Thus, direct NIR photoredox catalysis with a metal-free photocatalyst is a long-term goal in solar energy use. Notably, in the preparation of this paper, cyanines were reported to conduct NIR organic photoredox catalysis, constituting the only one example of metal-free NIR photoredox catalyst [13].

In other hand, iodinated BODIPY derivatives are well-known as metal-free photocatalysts for visible-light photoredox catalysis because of their strong light absorption ability (molar extinction coefficient of 10^5^ or more), high triplet quantum yield (>90%), long triplet excited state lifetime (>2 µs), robust photostability, and easily tailorable structure [14,15]. However, there are no reports of NIR-activated BODIPY as photocatalysts. Recently, carbazole-substituted iodinated BODIPY (BDP) was utilized as an efficient photosensitizer for ultralow-power NIR-triggered photodynamic therapy thanks to its intense absorption in the NIR region and its remarkably high singlet oxygen quantum yield (67%) [16]. Considering the iodinated BODIPY core and the solubility-improving long chain of BDP, we anticipated that BDP can be applied as an efficient metal-free photocatalyst via direct NIR light utilization.

Herein we report the successful utilization of NIR light (720 nm LED) as the energy source for BDP-catalyzed photooxidation (Figure 1). After being excited by NIR light, BDP undergoes the intersystem crossing (ISC) process to reach its triplet-excited state (^3^BDP^*^); this in turn can transfer energy to oxygen, forming the singlet oxygen (^1^O_2_), or conduct electron transfer process to finally generate the superoxide anion (O_2_^•―^). These generated reactive oxygen species are the terminal oxidant to achieve efficient oxidation of benzylamine derivatives, sulfides, and aryl boric acids. Notably, compared to the state-of-art blue-light photocatalyst Ru(bpy)_3_Cl_2_, this NIR photocatalyst BDP exhibits solvent independence and better scalability in photooxidation, possibly benefiting from the neutral molecular structure and penetration power of NIR light. Furthermore, we demonstrate the wide applicability of NIR-driven photooxidation with BDP through aryl boric acid–involving prodrug activation as well as the combination with Cu catalysis for carbon–carbon cross coupling.

## 2. Results and Discussion

BDP was synthesized according to the reported procedure [16], and the photophysical properties of BDP were investigated in dichloromethane (DCM). As shown in Appendix A, the absorption peak of BDP located at 709 nm with a high molar extinction coefficient (9.5 × 10^4^ M^−1^cm^−1^). In addition, no obvious photobleaching of BDP was detected after 3 h of NIR illumination (720 nm LED, 20 mW/cm^2^) (Appendix A), verifying its robust photostability. The fluorescence emission peak of BDP located at 750 nm with a quantum yield as low as 4% since the heavy atom effect of iodine promoted the transition from ^1^BDP^*^ to ^3^BDP^*^ [15,16]. Importantly, the singlet oxygen quantum yield of BDP was measured to be 65% at 710 nm using the established method [15].

In view of the high ^1^O_2_ production efficiency of BDP, BDP was then used as the photocatalyst for the benzylamine coupling reaction under NIR light illumination. The photooxidation coupling of benzylamine with ^1^O_2_ as the terminal oxidant was an important probe reaction to afford Schiff bases [17,18,19,20], which are useful building blocks in the synthesis of fine chemicals, functional materials, and useful drugs [21,22,23]. Conventional photocatalysts such as g-C_3_N_4_, TiO_2_, Ir(ppy)_3_, and [Ru(bpy)_3_]^2+^ were all applied in the photooxidation of benzylamine (**1a**), utilizing visible light or UV as the excitation source [17,18,19,20]. With extra-low BDP loading (~0.06 mol%) and mild NIR irradiation (720 nm, 20 mW/cm^2^), the quantitative conversion of benzylamine to a Schiff base (**2a**) was observed after 2 h reaction in DCM (See Appendix A for detailed reaction setup). No product was detectable in the absence of light, BDP, or oxygen, indicating that NIR illumination, BDP, and oxygen were all required for this reaction to proceed (Appendix A). Additionally, the effect of solvent polarity on this NIR-driven BDP-catalyzed benzylamine coupling was explored. High conversions of benzylamine were obtained with various solvents such as acetonitrile (CH_3_CN, a highly polar aprotic solvent), methanol (MeOH, a highly polar protic solvent), and the medium polar solvents DCM and ethyl acetate (EtOAc), suggesting that the polarity of the solvent has an insignificant effect on BDP-catalyzed NIR-driven photocatalysis (Appendix A). By contrast, Ru(bpy)_3_Cl_2_ exhibited strong dependence of the solvent for benzylamine photooxidation. In CH_3_CN, Ru(bpy)_3_Cl_2_ enabled the formation of a Schiff base with a conversion of 100% whereas only 5% conversion was obtained in DCM (Appendix A). This solvent dependence of Ru(bpy)_3_Cl_2_ may result from the sensitivity of the chloride counter-anion [24], which in turn reflected the robustness of the organic-neutral NIR photocatalyst, BDP.

The scale-up photooxidation of benzylamine with the combination of BDP and NIR light or Ru(bpy)_3_Cl_2_ and blue light was then explored. The amounts of the substrate benzylamine and photocatalysts were enlarged by 20 times for a 20 mL reaction. As shown in Figure 2, Ru(bpy)_3_Cl_2_ gave a sharp decline in conversion rate from 100% at 1 mL to 38% at 20 mL. In sharp contrast, the high conversion rate of 88% was still achieved with BDP as photocatalyst under NIR irradiation in 20 mL reaction. This result clearly demonstrated that NIR light can penetrate deeper into a reaction solution than visible light, and thus NIR-triggered photoredox catalysis is more suitable and efficient for large-scale or industrial application.

To explore the substrate scope of this NIR light–driven BDP-catalyzed aerobic oxidation, a variety of benzylamine derivatives were investigated. Electron-rich amines such as 4-methoxybenzylamine (Table 1, entry 2, **2b**) and electron-deficient amines substituted by halogens or trifluoromethyl groups (Table 1, entries 3–7) were all efficiently converted to the corresponding Schiff base, indicating that the electronic nature of substrates does not impact NIR-driven photooxidation. The nearly identical photoconversion rates of *ortho*- (**1d**), *meta*- (**1e**), and *para*-chloro (**1c**)–substituted benzylamines demonstrated that the position of the substituents had little impact on this NIR-driven BDP-catalysis (Table 1, entries 3–5). Moreover, the good conversion (77%) of benzylamine with steric hindrance (Table 1, entry 8) showed that steric hindrance cannot inhibit efficient photocoupling. Secondary amine substrates such as tetrahydroisoquinoline and dibenzylamine can also be transformed to the corresponding Schiff base by a BDP photocatalyst with a conversion of 59% and 100%, respectively (Table 1, entries 9–10). The above experimental results verified that BDP was an excellent NIR photocatalyst to efficiently promote the oxidative coupling of benzylamine derivatives with broad substrate scope.

The mechanism investigation of this NIR-driven BDP-catalyzed benzylamine coupling was then performed. In the presence of a singlet oxygen quencher (triethylenediamine, DABCO) [25], the generation of the Schiff base was negligible, indicating that singlet oxygen played an important role in this photoredox catalysis (Appendix A). In contrast, a good isolated yield of 84% was still obtained after adding high concentration of benzoquinone (BQ) [25], the trapping agent of superoxide anion, which ruled out the dominative role of superoxide anion in this NIR-photocatalytic system. In addition, the reaction rate in deuterated chloroform (CDCl_3_) was nearly twice that in DCM (Appendix A). Since deuterated solvents were reported to stabilize ^1^O_2_ [26], this enhanced reaction rate in CDCl_3_ further supported the assumption that ^1^O_2_ was the key species for this BDP-catalyzed NIR photooxidation. To confirm this proposal, the ^1^O_2_ generation in the presence of BDP was measured by using 1,3-diphenylbenzofuran (DPBF) as the ^1^O_2_ indicator [15]. Under the irradiation of NIR light (720 nm, 5 mW/cm^2^), the absorption of DPBF was significantly reduced in the presence of BDP (Appendix A), suggesting that BDP could trigger the generation of ^1^O_2_ with NIR light irradiation. Combined with the above experimental results, the proposed mechanism of this BDP-catalyzed NIR-driven oxidation of amines was outlined (Appendix A). BDP was firstly excited by NIR light to reach its singlet excited state (^1^[BDP]^*^); then it underwent the intersystem crossing (ISC) process to generate the triplet excited state ^3^[BDP]^*^ [15]. Singlet oxygen (^1^O_2_) was generated via triplet–triplet energy transfer from ^3^[BDP]^*^ to molecular oxygen. The substrate benzylamine was firstly oxidized by ^1^O_2_ to produce hydrogen peroxide and the intermediate imine, which further condensed with the second molecule of benzylamine to yield the final product, the Schiff base.

As shown in Figure 3, this NIR photooxidation of benzylamine could be further combined with copper catalysis (using copper trifluoroacetate, Cu(OTf)_2_, as catalyst), where the Schiff base **2a**, the in situ generated product of NIR photocatalysis, reacted with 4-*tert*-butylphenylacetylene in toluene to give an alkyne-substituted secondary amine with an isolated yield of 80% [27]. This result demonstrated that NIR light–driven photocatalysis can be merged with other catalytic systems in a tandem manner to access sophisticated functional scaffolds [28].

Besides the photooxidation of benzylamine, the photooxidation of sulfides via the ^1^O_2_ pathway was also of fundamental importance since the targeted sulfoxide motifs existed widely in organic intermediates, pharmaceutical molecules, and organic semiconductors [29,30,31,32]. BDP was then applied to NIR light–driven photooxidation of sulfides in view of its excellent singlet oxygen generation ability upon NIR light. As shown in Table 2, different sulfide derivatives, such as thioanisole (**3a**), benzyl methyl sulfide (**3b**), dibenzyl sulfide (**3c**), and 4-methoxyphenyl methyl sulfide (**3d**), were all smoothly oxidized to the corresponding sulfoxide with nearly quantitative conversions. A control experiment in the presence of DABCO led to inferior performance (11% yield), reflecting the dominative role of ^1^O_2_. This result further verified that BDP was an efficient NIR photocatalyst that can generate ^1^O_2_ as the terminal oxidant upon NIR light irradiation [33].

The triplet excited state of the photosensitizer was not only capable of sensitizing other molecules through an energy transfer pathway but can also participate in an electron transfer process to activate a substrate [15]. To test the feasibility of NIR-initiated ^3^BDP^*^ in the electron transfer process, the aerobic oxidation of aryl boronic acids (**5a**–**5f**) was employed as the probe reaction (Table 3) [34,35]. The reactions were completed within 4 h to give the formation of corresponding substituted phenols, with satisfactory yields of 85–95 %. Compared to the visible light catalytic system using Ru(bpy)_3_Cl_2_, the BDP-involved NIR photocatalysis furnished this reaction within much shorter time scopes [34]. Through comparative studies (Appendix A), we confirmed that BDP, the electron donor triethylamine (TEA), photoirradiation, and oxygen were all indispensable for this photocatalytic oxidation. In addition, in the presence of benzoquinone (quencher of O_2_^•―^), the yield of the product dramatically dropped from 90% to less than 10%, which implied the key oxidant role of O_2_^•―^ in the oxidation of boronic acids, showcasing the capability of this BDP-driven NIR photocatalysis in electron transfer–involved applications [25].

Utilizing the successful photooxidation of aryl boronic acid, we further explored the NIR-irradiated deprotection of prodrug in the presence of BDP. In comparison to the established deprotection approaches with acid/base-sensitive or redox-sensitive agents, the photolytic activation of prodrug emerged as a traceless and green alternative to allow deprotection under light illumination [36,37]. In the presence of BDP and NIR irradiation, arylborate moieties, the protecting groups of carboxylate drugs, were firstly oxidized to phenols, which then underwent photolysis to release the corresponding pharmaceutical scaffolds such as Naproxen (**8a**) and Indomethacin (**8b**), in the yields of 74% and 67%, respectively (Figure 4) [38,39,40]. These results revealed the potential of our NIR light–driven photolytic deprotection strategy in a wide array of applications related to the synthesis of functional photocaged small molecules [41,42].

## 3. Materials and Methods

### 3.1. Measurement of Singlet Oxygen Generation Using DPBF as the Indicator

The mixture of BDP (10 µM) and DPBF, of which the concentration was set to make the absorbance at 414 nm to be 1.0, was irradiated with 720 nm LED (5.0 mW/cm^2^). The absorption of the obtained solution was monitored every 20 s. The decrease in absorbance intensity of DPBF at 414 nm suggested that singlet oxygen was generated.

### 3.2. Photooxidation of Benzylamine Catalyzed by BDP with NIR Light Illumination

The mixture of benzylamine (0.275 mmol, 30 µL), BDP (1.6 × 10^−4^ mmol, 0.2 mg) in 1 mL dichloromethane was irradiated by 720 nm LED (20 mW/cm^2^). After the reaction, the solvent was removed under reduced pressure. Then 0.6 mL deuterated chloroform was added for the measurement of conversion via ^1^H NMR. For the benzylamine (NH_2_-**CH_2_**), the characteristic proton chemical shift located at 3.82 ppm; meanwhile, the unique chemical shift of the Schiff base product (**CH_2_**-N=CH) located at 4.82 ppm. Based on the integrated area of these two peaks, the product conversion can be calculated.

### 3.3. The Setup of the Large-Scale Reactions 

For the large-scale reaction under NIR light and blue light, the reactions were operated with a 25 mL vial under light irradiation of 20 mW/cm^2^. The mixture of benzylamine (5.5 mmol, 600 µL) and BDP (3.2 × 10^−3^ mmol, 4.0 mg) in 20 mL dichloromethane was irradiated by NIR light of 720 nm under air atmosphere. Meanwhile, the mixture of benzylamine (5.5 mmol, 600 µL) and Ru(bpy)_3_Cl_2_·6H_2_O (3.2 × 10^−3^ mmol, 2.4 mg) in 20 mL acetonitrile was irradiated by blue light of 455 nm. After the reaction, the solvent was removed under reduced pressure. Then 0.6 mL deuterated chloroform was added for the measurement of conversion via ^1^H NMR. For the benzylamine (NH_2_-**CH_2_**), the characteristic proton chemical shift was located at 3.82 ppm, and the unique chemical shift of the Schiff base product (**CH_2_**-N=CH) was located at 4.82 ppm. Based on the integrated area of these two peaks, the product conversion can be calculated.

### 3.4. Photooxidation of Sulfides Catalyzed by BDP with NIR-Light Illumination 

The mixture of sulfides (0.20 mmol) and BDP (1.6 × 10^−4^ mmol, 0.2 mg) in 2 mL DCM/methanol (*v*/*v* = 1/1) was irradiated by 720 nm LED (20 mW/cm^2^) under air atmosphere. After the reaction, the solvent was removed under reduced pressure. Then 0.6 mL deuterated chloroform was added for the measurement of conversion via ^1^H NMR. For the sulfide (S-**CH_2_** or S**-CH_3_**), the characteristic proton chemical shift was located near 2.40 ppm, while the unique chemical shift of the sulfoxide product (**CH_2_**-S=O or **CH_3_**-S=O) was located near 2.72 ppm. Based on the integrated area of these two peaks, the conversion can be calculated.

### 3.5. Photooxidation of Phenylboronic Acids Catalyzed by BDP with NIR-Light Illumination

The mixture of phenylboronic acids (0.20 mmol), triethylamine (TEA) (50 µL), and BDP (1.6 × 10^−4^ mmol, 0.2 mg) in 2 mL DCM/methanol (*v*/*v* = 1/1) was irradiated by 720 nm LED (20 mW/cm^2^) under air atmosphere. After the reaction, water was added. The aqueous mixture was then extracted with ethyl acetate (EtOAc) following which the ethyl acetate layer was dried with anhydrous Na_2_SO_4_ and concentrated. This was followed by column chromatography over silica gel to yield the product eluent EtOAc/hexane = 3/1, *v*/*v*.

### 3.6. Turnover Number (TON) and Turnover Frequency (TOF) Calculation

Based on the ^1^H NMR data, the conversion of the substrate can be calculated and then the amount of the product can be determined.
TON = mole of product per mole of catalyst(1)
TOF = TON/irradiation time (min)(2)

### 3.7. Tandem Reaction Catalyzed by BDP and Copper Salt 

After two hours reaction of benzylamine (0.825 mmol) in the presence of the BDP (4.8 × 10^−4^ mmol) and 720 nm light illumination (20 mW/cm^2^) under air atmosphere, the solvent was removed under a vacuum. Then the reaction mixture containing imine product, a toluene solution containing 4.1 × 10^−2^ mmol Cu(OTf)_2_, 4.1 × 10^−2^ mmol Na_2_SO_4_ (10 mol%), and 100 µL 4-tert-butylphenylacetylene, was transferred to a dried Schleck tube under nitrogen atmosphere. After reacting at 90 °C overnight, the final product of this tandem reaction was isolated by flash column chromatography on silica gel with a mixed eluent of hexane and ethyl acetate (10:1, *v*/*v*).^1^H NMR (500 MHz, CDCl_3_): *δ* = 8.36 (s, 1H), 7.78–7.76 (m, 2H), 7.43–7.39 (m, 5H), 7.32–7.29 (m, 5H), 7.24–7.22 (m, 1H), 4.81 (s, 2H), 3.01 (s, 1H), 1.29 (s, 9H).

## 4. Conclusions

In summary, we employed a metal-free BODIPY-derivative, BDP, as the NIR light photocatalyst to achieve the efficient oxidation of benzylamine derivatives, sulfides, and aryl boronic acids to yield chemicals with synthetic importance and added value. The strong absorption of NIR light of BDP enabled the direct utilization of low-power NIR irradiation (720 nm LED, 20 mW/cm^2^) for photooxidation, which exhibited deeper penetration depth and higher efficiency in large-volume reactions than the established visible- light photocatalytic protocols. Mechanism investigation showed that BDP can effectively produce singlet oxygen via energy transfer or generate superoxide anion via electron transfer upon NIR illumination. More importantly, BDP-triggered NIR photooxidation can be further combined with Cu catalysis to yield alkyne-substituted secondary amine derivatives, or applied in photolytic prodrug activation of Naproxen-borate and Indomethacin-borate, demonstrating the infinite potential of NIR photocatalysis in sophisticated organic synthesis. This work showcased the capability of utilizing NIR light by organic dyes to forge value-added chemicals, which will benefit the development of organic synthesis, material design, and solar energy use.

## Figures and Tables

**Figure 1 molecules-27-04047-f001:**
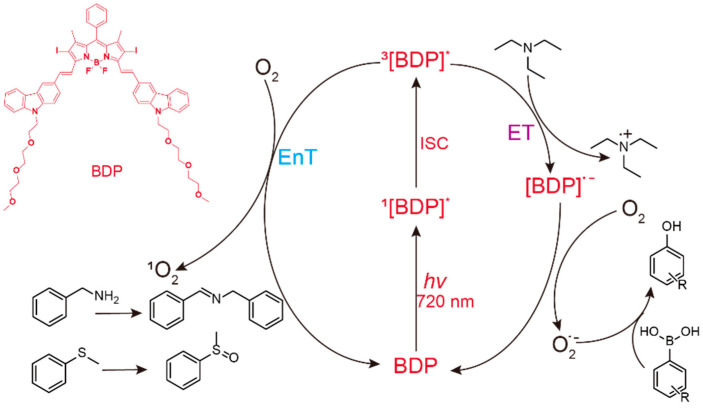
Mechanism of NIR-driven BDP-catalyzed photooxidation.

**Figure 2 molecules-27-04047-f002:**
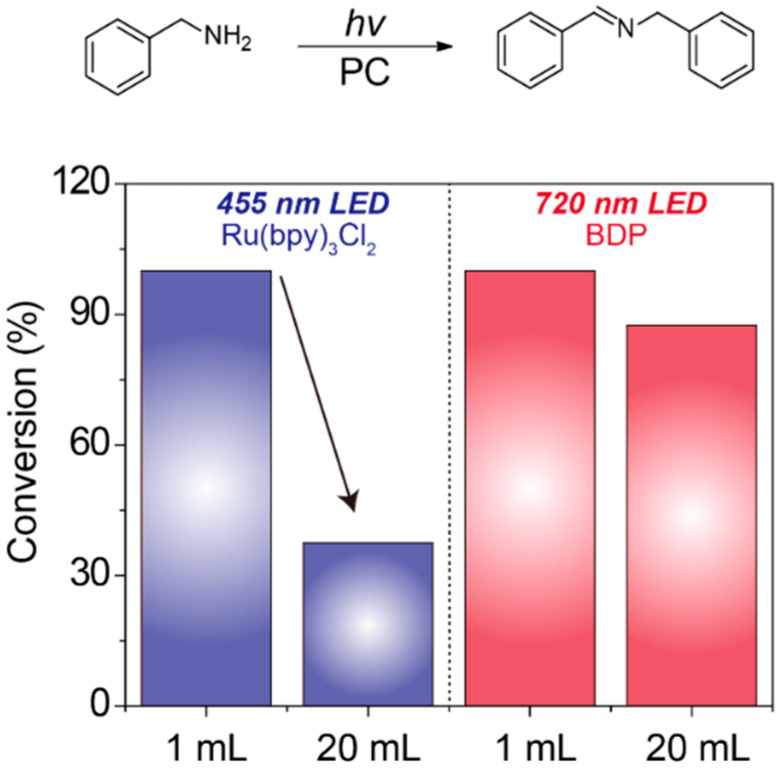
Comparative photocatalytic performance of Ru(bpy)_3_Cl_2_ and BDP for benzylamine coupling in different reaction volumes.

**Figure 3 molecules-27-04047-f003:**
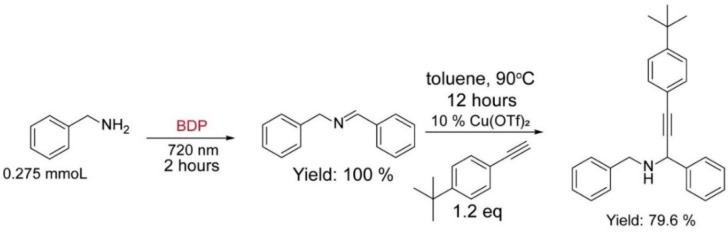
The tandem synthesis of the alkyne-substituted secondary amine via the combination of NIR light–driven BDP photocatalysis with copper catalysis.

**Figure 4 molecules-27-04047-f004:**
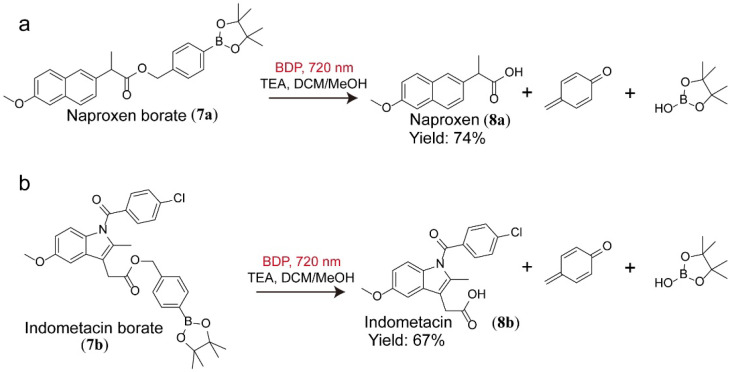
NIR light–driven photolytic activation of the prodrug naproxen borate (**a**) and indometacin borate (**b**) with BDP as the photocatalyst.

**Table 1 molecules-27-04047-t001:** Oxidation of various amines using BDP as the photocatalyst under NIR illumination *^a^*.

Entry	Substrate	Product	Conversion *^b^*	TON *^c^*	TOF (min^−1^) *^d^*
1	** 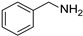 1a**	** 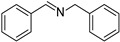 2a**	100	8.59 × 10^2^	7.16
2	** 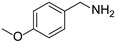 1b**	** 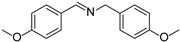 2b**	100	8.59 × 10^2^	7.16
3	** 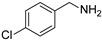 1c**	** 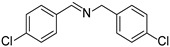 2c**	95.6	8.22 × 10^2^	6.85
4	** 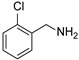 1d**	** 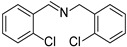 2d**	95.2	8.18 × 10^2^	6.82
5	** 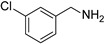 1e**	** 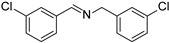 2e**	100	8.59 × 10^2^	7.16
6	** 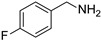 1f**	** 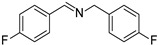 2f**	100	8.59 × 10^2^	7.16
7	** 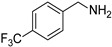 1g**	** 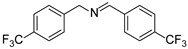 2g**	100	8.59 × 10^2^	7.16
8 *^e^*	** 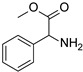 1h**	** 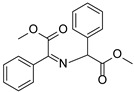 2h**	76.9	6.61 × 10^2^	5.51
9	** 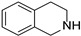 1i**	** 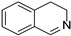 2i**	58.8	5.05 × 10^2^	4.21
10	** 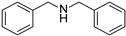 1j**	** 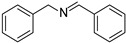 2j**	100	8.59 × 10^2^	7.16

*^a^* Reaction conditions: amine (0.275 mmol), BDP (1.6 × 10^−4^ mmol, 0.2 mg), DCM (1 mL), 720 nm LED (20 mW/cm^2^), under air atmosphere, 2 h, room temperature. *^b^* Conversion determined by ^1^H NMR. *^c^* Turnover number (TON) value was calculated as mole of amine converted per mol of BDP. *^d^* Turnover frequency (TOF) was equal to TON divided by irradiation time. *^e^* In methanol.

**Table 2 molecules-27-04047-t002:** Oxidation of various sulfides using BDP as the photocatalyst under NIR illumination *^a^*.

Entry	Substrate	Product	Conversion *^b^*	TON *^c^*	TOF (min^−1^) *^d^*
1	** 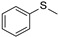 3a**	** 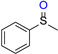 4a**	100	430	3.6
2	** 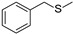 3b**	** 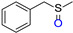 4b**	100	430	3.6
3	** 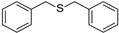 3c**	** 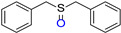 4c**	96	413	3.5
4	** 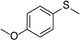 3d**	** 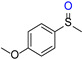 4d**	90	387	3.2

*^a^* Reaction condition: sulfide (0.20 mmol), BDP (1.6 × 10^−4^ mmol, 0.2 mg), DCM/methanol (*v*/*v* = 1/1, 2 mL), 720 nm LED (20 mW/cm^2^), in air, 4 h, room temperature. *^b^* Conversion was determined by ^1^H NMR. *^c^* TON value was calculated as mole of sulfide converted per mol of BDP. *^d^* TOF means turnover frequency, which equates to TON divided by irradiation time.

**Table 3 molecules-27-04047-t003:** Oxidation of various phenylboronic acid derivatives using BDP as the photocatalyst under NIR illumination *^a^*.

Entry	Substrate	Product	Yield *^b^*	TON *^c^*	TOF (min^−1^) *^d^*
1	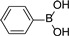 **5a**	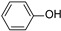 **6a**	85	365	3.1
2	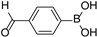 **5b**	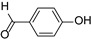 **6b**	95	408	3.4
3	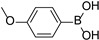 **5c**	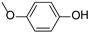 **6c**	90	387	3.2
4	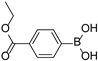 **5d**	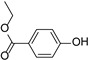 **6d**	87	374	3.1
5	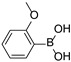 **5e**	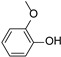 **6e**	94	404	3.4
6	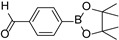 **5f**	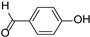 **6f**	95	408	3.4

*^a^* Reaction condition: phenylboronic acid derivatives (0.20 mmol), triethylamine (TEA) (50 µL), BDP (1.6 × 10^−4^ mmol, 0.2 mg), DCM/methanol (*v*/*v* = 1/1, 2 mL), 720 nm LED (20 mW/cm^2^), in air, 4 h, room temperature. *^b^* Isolated yield. *^c^* TON value was calculated as mole of phenylboronic acid converted per mol of BDP. *^d^* TOF means turnover frequency, which equates to TON divided by irradiation time.

## Data Availability

Not applicable.

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
