# Peer review of "Direct Utilization of Near-Infrared Light for Photooxidation with a Metal-Free Photocatalyst"

_molecules, 2022, doi:10.3390/molecules27134047_

Round 1
Reviewer 1 Report
This work by Duan and Zhang et al. reports several NIR-driven photoreactions including oxidation of benzylamine, sulfide and arylboronic acid using a metal-free boron-dipyrromethene (BODIPY)-derivative as photocatalyst. The NIR-driven photooxidation exhibited solvent independence and showed superior performance over conventional UV driven photoreactions when in scale-up experiments, due to the deeper penetration depth of NIR light. The substrate scope and mechanisms have also comprehensively investigated. Appropriate and solid experiments have been conducted in this work, I therefore recommend publication of this article, provided the few points and typos outlined below can be addressed by the authors to improve the manuscript.
1. This paper is suggested to be polished by an English native to correct several grammatical & spelling mistakes that are present in the text
P1, L33, “exhibits higher penetration depth through various media” should be “exhibits higher penetration depth in various media”
P1, L35, “thus benefiting the photon absorption in scale-up reactions” is hard to understand, please consider revising
P2, L53, “photoredox catalysis” should be “photoredox catalyst”
2. For the scale-up experiment of the photooxidation of benzylamine, using 455 nm and 720 nm LED as the excitation source, the authors observed sharp decreased of the conversion rate when using 455 nm LED, the light penetration capability will surely influence the conversion, but other experimental setup will also influence the photoreaction results, such as irradiation surface, light power, stirring of the solution, etc. thus, a detailed experimental setup description was highly suggested to be added, a setup picture is also welcome.
3. some relevant work about the photocatalysis are suggested to be added, such as “ACS Appl Mater Interfaces 2022, 14, 18, 21453–21460 and Org. Lett. 2018, 20, 1680-1683.”
Author Response
Comments and Suggestions for Authors
This work by Duan and Zhang et al. reports several NIR-driven photoreactions including oxidation of benzylamine, sulfide and arylboronic acid using a metal-free boron-dipyrromethene (BODIPY)-derivative as photocatalyst. The NIR-driven photooxidation exhibited solvent independence and showed superior performance over conventional UV driven photoreactions when in scale-up experiments, due to the deeper penetration depth of NIR light. The substrate scope and mechanisms have also comprehensively investigated. Appropriate and solid experiments have been conducted in this work, I therefore recommend publication of this article, provided the few points and typos outlined below can be addressed by the authors to improve the manuscript.
Response: We highly appreciated the reviewer’s recognition for our effort to elucidate the benefit of BODIPY as the metal-free NIR photocatalyst. We tried our best to revise and improve the manuscript. The point-to-point responses are listed as below.
- This paper is suggested to be polished by an English native to correct several grammatical & spelling mistakes that are present in the text
P1, L33, “exhibits higher penetration depth through various media” should be “exhibits higher penetration depth in various media”
P1, L35, “thus benefiting the photon absorption in scale-up reactions” is hard to understand, please consider revising
P2, L53, “photoredox catalysis” should be “photoredox catalyst”
Response: Thanks to the reviewer’s kindly suggestion and detailed review. We have corrected the above-mentioned issues and thoroughly revised this paper with the help of native-English speakers, and the corresponding revised texts were highlighted.
- For the scale-up experiment of the photooxidation of benzylamine, using 455 nm and 720 nm LED as the excitation source, the authors observed sharp decreased of the conversion rate when using 455 nm LED, the light penetration capability will surely influence the conversion, but other experimental setup will also influence the photoreaction results, such as irradiation surface, light power, stirring of the solution, etc. thus, a detailed experimental setup description was highly suggested to be added, a setup picture is also welcome.
Response: Thanks to the suggestion of reviewer. We have added the detailed experimental setup description in Materials and Methods section. The light power density and stirring speed of the solution remained constant for all photocatalysis cycles. Regarding the large-scale reaction and the small-scale reaction, the irradiation surfaces were indeed different but the changes were simultaneously occurred to the reactions under blue light or NIR light. Thus, the corresponding statement that the different response upon the increase of reaction size originates from the distinct penetrating power between NIR light and blue light, is reasonable.
- some relevant work about the photocatalysis are suggested to be added, such as “ACS Appl Mater Interfaces 2022, 14, 18, 21453–21460 and Org. Lett. 2018, 20, 1680-1683.”
Response: We appreciated the reviewer’s suggestive comments. The mentioned references are excellent examples for the utilization of longer wavelength light for photoredox catalysis, and those two references were added as Refs 6 and 7.

Reviewer 2 Report
The authors employ a BODIPY derivative as the catalyst for the NIR-induced photooxidation of some substrates. Thus, the oxidative coupling of benzylamines driving to imines, the oxidation of sulfides to sulfoxides and the oxidation of phenylboronic acid derivatives to phenols have been achieved. In addition, some applications of this methodology are shown.
The work is interesting, as very few examples of potentially useful NIR-induced photooxidations, using metal-free photocatalysts can be found. Therefore, the manuscript could be suitable for publication. However, some additions/clarifications should be made before being accepted, as follows:
1) A recent review by Cormier and Goddard about near-infrared photoredox catalysis for organic synthesis (Org. Chem. Front. 2021, 8, 6783) should be included in the introduction (line 40).
2) The explanation presented in the Materials and Methods (lines 232-236) shows that the obtained yields are conversions, which should be indicated in the tables. In fact, the authors present the 77% yield of Table 1, entry 8 as a “good conversion” (line 137). Of course, if the conversion is close to 100% and no other impurity is presented by NMR, and no work-up exists, this can be considered quite close to the yield. However, in the case of lower conversions (for example, 76.8 or 58.8%, Table 1, entries 8 and 9) and with impurities present in the NMR spectra, it should be better to use quantitative NMR with an internal standard (if chromatography is not possible). These values can be added in the table, for example, in parentheses beside the conversion.
3) The scheme of the tandem photooxidation of benzylamine combined with copper catalysis should be moved from the supporting information to the manuscript for clarity, as the reaction is mentioned in the abstract, and even the experimental procedure is included in the Materials and Methods.
4) The experimental procedures for the oxidation of sulfides and phenylboronic acids should be included in the Materials and Methods section.
Author Response
Comments and Suggestions for Authors
The authors employ a BODIPY derivative as the catalyst for the NIR-induced photooxidation of some substrates. Thus, the oxidative coupling of benzylamines driving to imines, the oxidation of sulfides to sulfoxides and the oxidation of phenylboronic acid derivatives to phenols have been achieved. In addition, some applications of this methodology are shown.
The work is interesting, as very few examples of potentially useful NIR-induced photooxidations, using metal-free photocatalysts can be found. Therefore, the manuscript could be suitable for publication. However, some additions/clarifications should be made before being accepted, as follows:
Response: We highly appreciated the positive comments of reviewer and the revision chance that provided, and the point-to-point responses to the reviewer’s concerns are listed as below.
1) A recent review by Cormier and Goddard about near-infrared photoredox catalysis for organic synthesis (Org. Chem. Front. 2021, 8, 6783) should be included in the introduction (line 40).
Response: Thanks for the reviewer’s kindly suggestion. The mentioned reference is an updated review for NIR photoredox catalysis, which was added as Ref 10.
2) The explanation presented in the Materials and Methods (lines 232-236) shows that the obtained yields are conversions, which should be indicated in the tables. In fact, the authors present the 77% yield of Table 1, entry 8 as a “good conversion” (line 137). Of course, if the conversion is close to 100% and no other impurity is presented by NMR, and no work-up exists, this can be considered quite close to the yield. However, in the case of lower conversions (for example, 76.8 or 58.8%, Table 1, entries 8 and 9) and with impurities present in the NMR spectra, it should be better to use quantitative NMR with an internal standard (if chromatography is not possible). These values can be added in the table, for example, in parentheses beside the conversion.
Response: Thanks for the suggestive comments. We have changed the phrase of “yield” to “conversion” in the section of Materials and Methods to demonstrate the photocatalytic results more accurately. Within the 1H NMR spectra of crude reaction mixture, there are no other peaks than that of the substrate and the product listed in Table 1. This situation is not only the case for the substrate with conversions higher than 90%, but also found for entries 8 and 9 exhibiting inferior photooxidation performances. This high selectivity for the photooxidation of benzylamine derivatives has also been reported when using mpg-C3N4 (Angew. Chem. Int. Ed. 2011, 50, 657 –660) and MOF PCN-777 (Angew. Chem. Int. Ed. 2018, 57, 5379 –5383) as the photocatalyst.
3) The scheme of the tandem photooxidation of benzylamine combined with copper catalysis should be moved from the supporting information to the manuscript for clarity, as the reaction is mentioned in the abstract, and even the experimental procedure is included in the Materials and Methods.
Response: Thanks for the reviewer’s kindly suggestion. We have moved this Scheme of tandem photooxidation to the manuscript as Figure 3.
4) The experimental procedures for the oxidation of sulfides and phenylboronic acids should be included in the Materials and Methods section.
Response: Thanks for the nice suggestion. We have added the experimental procedures for the oxidation of sulfides and phenylboronic acids to the section of Materials and Methods.
